# Socialized Farmland Operation—An Institutional Interpretation of Farmland Scale Management

**Yiqing Su** [1,2], **Qiaoyuan Huang** [2], **Qi Meng** [2,*], **Liangzhen Zang** [3] and **Hua Xiao** [4]

1   Regional Social Governance Innovation Research Center, Guangxi University, Nanning 530004, China
2   School of Public Policy and Management, Guangxi University, Nanning 530004, China
3   Institute of Agricultural Economics and Development, Chinese Academy of Agricultural Sciences, Beijing 100081, China
4   Rural Revitalization Service Center of Pingyuan County, Dezhou 253100, China
*   Correspondence: mengqi@st.gxu.edu.cn

**Abstract:** Farmland scale management is an important approach for developing countries to ensure food security in the face of the COVID-19 pandemic. At present, the realization of farmland scale management through the path of farmland use rights trading encounters obstacles in practice; moreover, the new model of farmland scale management has rarely been systematically discussed. Considering the farmland trusteeship practice implemented in Shandong Province of China as the research case, this study discusses the essence and realization premise of the new farmland scale management model represented by farmland trusteeship based on case analysis. The conclusions are as follows. (1) The high cost generated from farmland scale management is the main obstacle to realize this model. (2) The process of realizing farmland scale management through farmland trusteeship is actually the process of meeting the requirements of the socialization of farmland use, the socialization of the farmland management process, and the socialization of farmland output. Thus, in the context of the existence of a large number of small and scattered farmers in China, the socialized farmland operation is the essence of farmland scale management. (3) Effective collective action is the premise of realizing socialized farmland operation. Undeniably, a lot more systematic explorations are further demanded to strengthen the irrigation management and infrastructures, promote and ensure stable village leadership, and comprehensively improve the ability of rural collective action to ensure the further strengthening of socialized farmland operation so as to realize stable farmland scale management, which will be pursued in the future.

**Keywords:** farmland scale management; farmland use rights trading; farmland trusteeship; farmland transfer; China

## 1. Introduction

The COVID-19 pandemic is posing a serious threat to global food security through a variety of transmission mechanisms [1–3]. With the continuous progressing of the COVID-19 pandemic, shortages of agricultural labor created by restrictions on movements of people and the rise of food prices caused by restrictions on international trade have led to chaos in agricultural and food markets, leading to inadequate food supply and the disruption of food supply chains. Consequently, the world is falling into poverty, hunger, and potential food security crises [4–7]. Therefore, it is, in particular, important for developing countries to develop new strategies for the expansion of local food production in order to promote sustainable food security, reduce livelihood risks for farmers, and enhance the resilience of food systems in the current context [8,9].

As the most populous country in the world, China's cultivated land area accounts for only 9% of the global cultivated land [10]; however, it feeds nearly one-quarter of the world's population. Relevant data show that global food production in 2019 was 2722 billion kg, of which China produced 664 billion kg, accounting for nearly one-quarter

of global food production [7]. However, still, many factors constrain the national food security and agricultural development in China. Among them, the lack of grain-planting subjects, the expansion of off-farm employment, the change of grain production structure, and the outflow of rural labor force caused by the low efficiency of rural land production, idle land, and abandonment, have left a lot of hidden dangers to China's rural sustainable development and national food security [11–13]. Therefore, under the impact of COVID-19, how to continuously promote farmland scale management strategies to overcome the negative impact of idle land, abandoned land, and low land production efficiency on China's food security has become an important issue determining China's national security at present and in the future.

In order to overcome the long-standing problems of low agricultural production efficiency caused by land fragmentation, and land idling caused by the outflow of labor force, the Chinese government put forward a systematic farmland use rights trading (FURT) (also named as "farmland transfer") policy in 2002 [14,15]. The policy states that the farmers who do not want to farm their land can lease their farmland use rights to others, and those who want to farm larger land can achieve scaled, intensive operation of the land by renting in others' farmland use rights, thus acquiring scale economy. This leads to the alleviation of the land fragmentation problem under the premise of the existence of a large number of small and scattered farmers as well as an improvement in land production efficiency and economic income [16–18]. However, in the process of promoting the FURT, China has gradually encountered a series of problems. For example, the relevant laws and systems of participation of farmers in FURT are not perfect, farmers' rights and interests become damaged, and the market information of FURT is asymmetrical [19]. As a result, China has gradually been unable to achieve a higher level of farmland scale management through FURT in recent years, which has become a huge obstacle in the process of ensuring national food security [20].

In cases where the FURT failed to improve farmland scale management, an attempt to achieve farmland scale management through providing productive services for farmland operators came to the forefront (Providing productive services for farmland operators means that agricultural service companies provide agricultural production services to farmers. In the land trusteeship model, agricultural service companies have advanced, large agricultural production equipment, such as a large rice transplanter, large combine harvester, grain dryer and so on. They also have relatively advanced agricultural green technology, they provide farming, pest control, field management, harvesting and other operational services to a number of farmers). More in detail, farmland trusteeship was adopted by the Chinese farmers as a specific way to promote farmland scale management by providing productive services for farmland operators. The emergence of this new farmland scale management pattern enabled us to further explore the following two issues and fulfill the following objectives: (1) to evaluate the essence of farmland scale management, as represented by the new farmland scale management model, and (2) to investigate the prerequisites in order to promote the formation of a new farmland scale management model such as farmland trusteeship. In order to fulfill the above-mentioned goals, case studies were conducted on farmland trusteeship in Shandong Province, China to analyze the essential characteristics of the current farmland utilization model with productive services used to drive scale management. Moreover, the aim was to explore the necessary conditions to realize this new scale management model from the perspective of collective action in order to provide useful exploration for developing countries to effectively cope up with national food security, agricultural production, and farmers' livelihood issues under the impact of the COVID-19 pandemic.

This study may provide new insight into how to achieve farmland scale management in developing countries, in particular, in those developing countries with a large number of small-scale farmers. On the one hand, this study discusses the essential characteristics of farmland scale management, and it provides a new theoretical approach to understand the problem of farmland scale management in countries with a large number of small-scale

farmers. On the other hand, based on China's land trusteeship practice, this study presents the concrete process of how to achieve farmland scale management through socialized farmland operation, namely collective actions in land use, land production process, and land output distribution. This new concept of socialized farmland operation summarizes the operational principle of sustainable farmland scale management for countries with a large number of small-scale farmers. Moreover, through the concept of collective action involved in the socialized farmland operation, this study provides policy enlightenment for people to explore how to deal with the impact of COVID-19 epidemic through cooperation and collective actions.

## 2. Literature Review

"Farmland scale management", also meaning "scale operation of farmland" or "farmland scale operation", is a type of farmland management mode that centralizes a certain amount of cultivated land; gives full play to the ability of various production factors; improves the land yield rate, labor productivity, and commodity rate of agricultural products; and also improves the economic benefits. Among developing countries, farmland scale management is regarded as the inevitable course to promote agricultural modernization, and FURT is considered as a solid approach to achieve this purpose. On the one hand, FURT may improve agricultural production efficiency, allowing for the achievement of the economies of scale [21]. On the other hand, FURT can not only increase farmers' income by improving off-farm employment and land rental [22] but also concentrate farmland use rights in the hands of capable agricultural producers and operators, thereby introducing new factors in agricultural production and attracting industrial and commercial capital investments [23].

However, the limitations of FURT also began to emerge during its continuous development. Scholars have observed that in rural areas of China, 2 million hectares of land for agricultural production are abandoned every year [24]. It indicates that the policy of FURT does not necessarily indicate the practical occurrence of farmland scale management. Some scholars have attributed this to the fact that the advancement of FURT is often hindered by both economic and social factors. On the one hand, the economic benefits of FURT are not aligned with theoretical expectations [25]. For the leasee in the FURT, the concentration of factors through FURT does not lead to scale effect and does not improve farmers' planting efficiency in reality. In addition, due to the rigidly increasing farmland rent, high prices of agricultural inputs, and risks coming from both extreme weather events and the markets, the profit of agricultural management participant becomes significantly reduced. Those farms with industrial and commercial capital formed through FURT exhibit a dilemma of progressively diminishing or even under deficit scale effects because of high labor costs and difficulty in supervision [26]. On the other hand, the social benefits stemming from FURT are also poor. The social security attribute of farmland has hampered FURT and contributed to farmers' attachment to farmland. Consequently, in areas where it is difficult to have off-farm employment, farmers' willingness to participate in FURT is generally not strong [27]. Moreover, the overextension of FURT in some places has caused industrial and commercial capital to exclude small farmers, who depend on farmland for survival, from farmland operation and management. As a result, the small farmers lost their basis for keeping a foothold in the countryside, thereby causing rural society to gradually lose its robustness [28]. Furthermore, industrial and commercial capital investments in the countryside, promoted by large-scale FURT, also caused the replacement of traditional rural relationships with economic relationships, which further damaged the trust and relationship networks among villagers. This led to the reduction in the people's sense of belonging to their villages and caused the governance of public affairs and commons in rural areas to be mired in difficulties [29].

In the current situation, characterized by the fact that the level of farmland scale management cannot be further improved through the FURT, an attempt to provide farmland productive services for farmers simultaneously emerged at the right moment. Furthermore,

the model of farmland trusteeship, which was explored by the Chinese farmers, is a specific way to promote farmland scale management by providing productive services for farmland. Farmland trusteeship indicates that, based on the fact that it is not necessary to trade farmland use rights, rural households entrust either a certain farmland production link, or several farmland productions links, or even all farmland production links in the farmland management process to agricultural cooperatives or agricultural service organizations for management. Recently, many scholars have investigated the willingness of farmers to participate in farmland trusteeship. First, they argued that the level of farmers' risk preference could affect their willingness to participate in farmland trusteeship [30]. Specifically, the farmland trusteeship entails higher risks than FURT with fixed rental revenue; therefore, the higher the risk preference level of farmers, the more inclined they become to choose the farmland trusteeship model [31]. Second, it was found that the number of family members who are involved in off-farm employment can also affect the willingness of farmers to participate in farmland trusteeship [32]. Third, aspects such as the general policies on farmland trusteeship, local market development, the professionalism of service cooperatives, and the leadership and skills of the grassroots organizations in villages also affect the willingness of farmers to participate in farmland trusteeship [33].

To summarize, existing studies have comprehensively investigated the benefits and advantages of FURT, strongly affirming its important role in promoting the realization of moderate farmland scale management as well as the development of agriculture in rural areas. Scholars have also discussed the limitations of FURT. They identified two aspects worthy of further exploration, i.e., in the search for new ways to promote farmland scale management by providing productive services for farmland, the two aspects are: the lack of an analysis of the essence of new model of farmland scale management, and the lack of an exploration of the prerequisites for forming the new model of farmland scale management.

## 3. Case Selection and Description

### 3.1. Case Selection

The case study method allows for the in-depth analysis of complex development processes and presents the relationships, structures, and mechanisms involved in the development of a phenomenon through a substantial description and systematic understanding of the case and to grasp the dynamic interaction processes and the situational context involved [34]. Notably, Shandong Province is a large agricultural province in China. According to *Bulletin of the first National Geographic Survey of Shandong Province (2017)*, plains account for 65.56% of the total area of this province, and according to *China Statistical Yearbook (2021)*, the cultivated land area is equal to 6.462 million hectares, and the total grain output reached 54.468 billion kilograms according to *Statistical Yearbook of Shandong Province (2021)*. Under the background of China's farmland institution of the household contract system, Shandong Province was characterized by the decentralization and fragmentation of farmland use rights. Moreover, the cultivated land area per capita was only 0.078 hectares, which posed severe challenges to the regional process of promoting farmland scale management. In order to address these challenges, Shandong Province pioneered a farmland scale management model characterized by farmland trusteeship, which proved to be more effective in solving the problem of farmland scale management in the case of farmland fragmentation. Since 2009, some grassroots cooperatives in Shandong Province have begun to explore the provision of productive services such as the supply of agricultural materials and the substitution of farming and planting for farmers, thereby leading to the gradual development of a farmland trusteeship model characterized by serving multiple farmland production links as the basic feature. As of 2020, the area of farmland served by the farmland trusteeship model in Shandong Province reached 3.926 million hectares, accounting for about 60.755% of the total cultivated land area in the province.

In this study, the villages in Wucheng County and Xiajin County of Dezhou City, Shandong Province, China were selected as the research site because of their representativeness, which is reflected in the following two aspects. First, the farmland in Dezhou is mainly

dominated by cultivated land, with an area of about 633,256 hectares according to *Dezhou city third land survey main data bulletin (2022)*, accounting for 77.40% of the total agricultural land area. However, the per capita cultivated land area is only 0.113 hectares. Since 2013, Dezhou City has started to provide farmland trusteeship services. These efforts have been valued and supported by the government, such that it has become a representative area for farmland trusteeship. Second, the villages in Dezhou City have implemented different forms of farmland trusteeship projects and FURT projects, thus providing good samples to summarize the essence of scale management represented by farmland trusteeship through case comparison.

*3.2. Data Collection*

A survey was conducted in Wucheng County and Xiajin County in Shandong Province, China from May to June 2018, while a return visit was paid in January 2022. Firsthand information and related data on the cases were obtained through field visits and semi-structured interviews. In each visit and survey, the research team conducted interviews with county government staff, village cadres, and villagers. The interviews mainly focused on the following three aspects: the development process and the current situation of farmland scale management; the situation of organizations for farmland trusteeship and FURT in the villages involved; and the operating modes of farmland trusteeship and FURT. These aspects helped summarize the operating modes of local farmland scale management and the conditions to effectively realize farmland scale management.

*3.3. Case Overview*

3.3.1. The XSD Village: A Basic Model of Farmland Trusteeship

XSD Village faced a major problem: a large number of laborers worked away from their hometown, and almost no one remained to cultivate land. Initially, XSD Village planned to realize farmland scale management through FURT; however, unfortunately, FURT was not implemented smoothly. First, large-scale farmers involved in the operation of large-scale farmlands needed to confer and negotiate with 8–10 farmers on average to transfer 6.67 hectares of land. Such a high time- and energy-consuming process discouraged them from renting more land. Second, farmers involved in operating large-scale farmlands were afraid of the greater economic risks, which was entailed in the fact that the large-scale machinery they bought could work only for two months a year. Consequently, the management of larger-scale farmland could lead to more idle production tools. Third, farmers who operated large-scale farmlands generally perceived that it was too difficult to supervise the work of hired laborers. In many cases, during the harvest season, and only after the wages were paid, they could realize that the cultivation work was not well performed. Therefore, those farmers were unwilling to expand the farmland scale. Moreover, households who rented out their farmland also found that after they traded their farmland use rights to large-scale farmers, the harvest was not as good as that obtained in the past when the farmland was managed by their own family. Therefore, farmers in the village were moderately reluctant to trade their farmland use rights to other farmers for cultivation.

Nevertheless, the problem of leaving land uncultivated, caused by the massive outflow of laborers, still needed to be solved. In this respect, the village collective of XSD Village introduced excellent varieties of wheat seed from a seed company in the city, and they persuaded the farmers from house to house to plant those wheat seeds. After realizing that the entire 38.67 hectares of land in the entire village were planted with the same variety of wheat seeds, through a series of consultations in the Villagers' congress, it was unanimously agreed that under the condition of not trading the farmland use rights, the village committee would be entrusted to uniformly purchase agricultural productive services on behalf of all the villagers for the 38.67 hectares of farmland in order to carry out farmland scale management. The fees to purchase agricultural productive services were charged from the farmers by the village committee and then uniformly paid to agricultural service

organizations, which carried out sowing, fertilizer application, pest control, and harvesting on the 38.67 hectares of farmland. In this process, the farmers were considered responsible for supervising the on-site production by the service organizations. For example, in relation to harvesting, each farming household assigned a representative to the field to supervise the work performed by the machinery operators; in contrast, those farmers who did not stay in the village usually entrusted their relatives or neighbors to supervise on their behalf. The wheat harvested in season was sold to cooperatives, enterprises, or markets at the farmers' own choice.

Figure 1 shows the basic model of farmland trusteeship in XSD Village. In this way, in the farmland trusteeship model, farmers obtained agricultural products; agricultural service organizations gained transaction profits; and the village committee established trust relationships with farmers through constant communication and interaction, thus winning political reputation. In case of XSD Village, the village committee only played the role of organizer and coordinator in realizing farmland scale management; however, the farmers' farmland was still managed by decentralized farming households. However, by the uniform selection of seeds and purchasing productive services, the village committee connected the farmers' small-scale management and realized farmland scale management under the condition of not processing FURT.

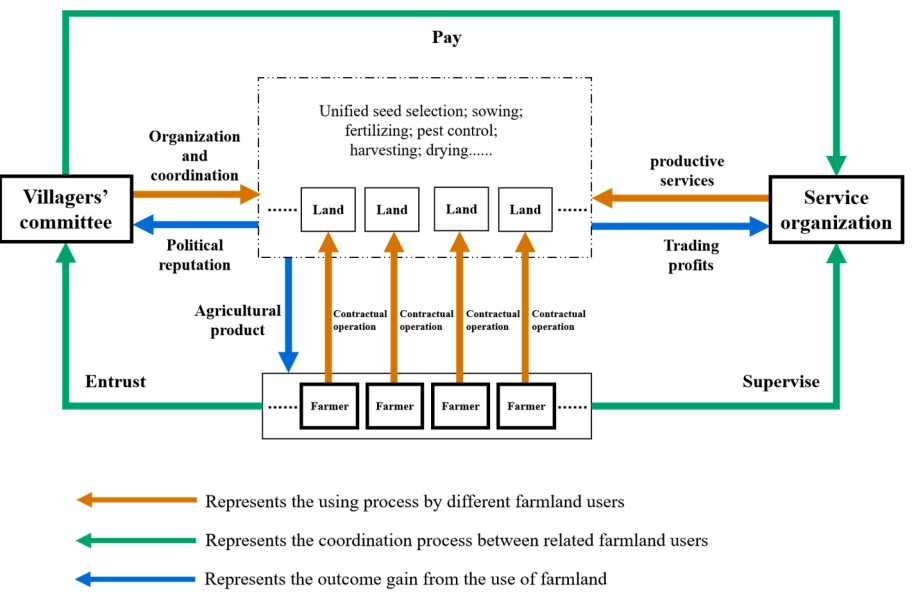

**Figure 1.** The basic model of farmland trusteeship in XSD Village.

### 3.3.2. The XXW Village: Farmland Trusteeship Combined with FURT through the Formation of Joint-Stock Cooperative

The XXW Village is located in Wucheng County, Dezhou City, Shandong Province, China. In this village, a basic model of the farmland joint-stock cooperative led by the Party branch was implemented in the farmland trusteeship model. The village has a population of approximately 1000 and 160 hectares of cultivated land, which were mainly planted with corn and wheat. Almost all the effective laborers worked away from their hometown. Therefore, it was urgent to solve the problem of who could cultivate the land. Initially, the plan of XXW Village was to encourage farmers to trade their farmland use rights to farmers who operated farmland scale management; however, the actual implementation of this method encountered significant obstacles. On the one hand, farmers were not willing to trade their farmland use rights, as they thought that they would face the risk of not having the possibility to take it back later, thereby putting their livelihood and survival in danger. On the other hand, large-scale farmers were reluctant to rent a large percentage of farmlands, because the management of large-scale farmland requires the reconstruction of water conservancy facilities and purchase of costlier large-scale machinery, thereby putting

a heavy burden on these farmers. In view of the obstacles in the promotion of FURT, it was expected that XXW Village should adopt new ideas to solve the problems of farmland operation and management.

In September 2017, the village cadres of XXW Village conducted door-to-door interviews and surveys to seek opinions from the villagers. Moreover, through consultations in Villagers' congress, the Party branch of the village signed an agreement for the pooling of farmland as shared with 102 villagers, with a total of 25.8 hectares of concentrated and contiguous farmland involved in FURT. Based on this, the first land joint-stock cooperative led by the Party branch was registered successfully in Wucheng County. Moreover, the cooperative elected an 11-member general assembly, a five-member council, and a supervisory board. The cooperative was set up to better administer the land trusteeship. Through consultation with farmers, the village cadres concentrated and connected the village land into big plain plots and supported the most basic farmland facilities, which provided convenience for the development of land trusteeship. In relation to the land joint-stock cooperative led by the Party branch, the council was responsible for purchasing agricultural productive services including sowing, fertilizer application, pest control, harvesting, and drying in a uniform manner on the concentrated big plots of farmland, by paying service fees to agricultural service organizations. During these processes, the supervisory board consisting of village representatives was responsible for the supervision of the management and business activities of the council, and the supervisors elected by villagers were responsible for supervising on-site agricultural production, while major issues were discussed jointly by the assembly of cooperative members. The land joint-stock cooperative distributed the dividends to the farmers joining the cooperative according to the annual planting cycles. The first-round dividend distribution was arranged after the sale of wheat. The cost of planting wheat in the corresponding season was calculated by considering the wheat yield standard of 6750 kg·ha$^{-1}$ as the basic output for the dividend distribution. Then, the wheat was sold at the current price of that year, and the earnings after subtracting the planting costs were distributed as dividends among the farmers joining the cooperative according to their shares. The second-round basic dividend distribution was arranged after the sale of corn. The cost of corn in the corresponding season was calculated, considering the corn yield standard of 8250 kg·ha$^{-1}$ as the basic output of the second dividend distribution. Then, the corn yield was sold at the market price of that year, and the earnings after subtracting the planting costs were distributed as dividends among the farmers who joined the cooperative. After distributing the two-round basic dividends, 50% of the remaining net profit of the cooperative at the end of each planting cycle was distributed again to the farmers joining the cooperative according to the capital stock certificate shares, while the other 50% was used for venture capitals, provident funds, and village collective income.

The farmland trusteeship model based on land joint-stock cooperative led by the Party branch allowed XXW Village to achieve good results. As of 2019, the number of farmer households joining the land-stock cooperative of XXW Village increased from 102 to 162, and an increasing number of farmer households participated in farmland trusteeship. Farmers joining the cooperative had more time to work away from their hometown, and they received more off-farm employment income based on ensuring their agricultural income. In relation to the village collective, the good operation of the land joint-stock cooperative led to the increase in the political reputation of the village cadres and won them the trust and affirmation from villagers, such that the village collective could have an annual collective income of more than 15,700 US dollars. Furthermore, agricultural service organizations not only obtained stable transaction profits but also established long-term cooperative partnerships with the village collective, thus achieving an all-win situation. Figure 2 shows the model of the farmland trusteeship cooperative led by the Party branch in XXW Village.

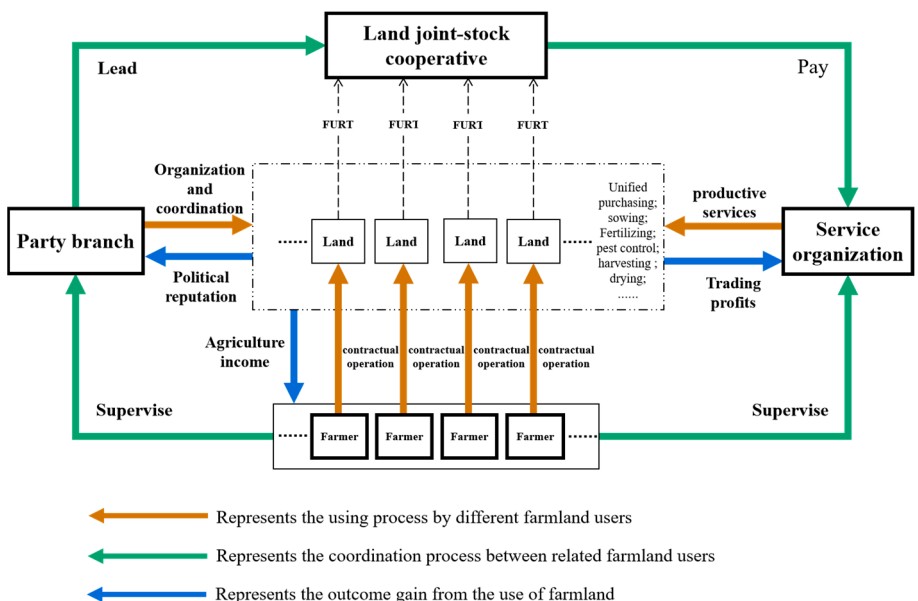

**Figure 2.** Model of the farmland trusteeship cooperative led by the Party branch in XXW Village.

### 3.3.3. The L Village: Retrogressive Farmland Scale Management

A contrasting case is represented for the L Village, not far from XSD Village and XXW Village, which did not adopt the farmland management model of the farmland trusteeship. The leaders of this village believed that the pooling of farmland as farmland trusteeship would be equivalent to putting the farmland again under unified collective management as in the 1960s. This was believed to entail the return of agricultural production and management to the "mess together" pattern that initially characterized the people's commune period, thereby causing damage to agricultural production. Based on an incomplete understanding of the modern farmland scale management, this village carried out farmland scale management by trading the farmland use rights to the farmers who were willing to operate large-scale farmlands.

At the beginning of the FURT in L Village, the entire village traded the use rights of the entire 113.33 hectares of farmland to a farmer for unified planting. This large-scale farmer paid rent to the farmers who traded their farmland use rights according to the agreement and no longer directly participated in farmland production and operation. However, due to the excessively large area of farmland to be managed, this large-scale farmer began to encounter the typical problems of high costs in production supervision and low production efficiency in the large-scale farmland production process. Therefore, after consultations in villagers' congress, the villagers of L Village agreed that the farmland rented by the large-scale farmer would be re-divided into 10 plots, and three other farmer households would be introduced to jointly manage the 113.33 hectares of farmland. Therefore, in L Village, the simple FURT resulted in the fact that the large-scale farmer constantly subdivided the land into relatively small-scale land for operation. Thus, the effect of farmland scale management was inevitably limited, and the scale management of farmland realized by using FURT was gradually managed by scattered households again.

## 4. Case Analysis

### 4.1. Extremely High Costs Hinder the Realization of Farmland Scale Management

The analyses of the cases of XSD Village and XXW Village indicate that in a country such as China, where the number of small-scale farmers is large, the property rights of farmland are fragmented, and the production is decentralized, the realization of farmland scale management is often accompanied by extremely high costs. This is specifically reflected by the following aspects.

The first is the extremely high coordination costs. Currently, in China, in order to gather 6.67 hectares of farmland for management, it is necessary to confer and negotiate with 8–10 households on average. Therefore, to meet the production capacity of a modern tractor with a standard full-load working area of 2000 hectares, it was required to negotiate with 2400–3000 households. The time and energy spent in negotiating with such a huge number of households are almost beyond the reach of any single producer and operating subject.

The second is the extremely high production costs. On the one hand, in the period when farmland was managed by scattered farmers, the infrastructure required for agricultural production was often provided through mutual consultation and cooperation among households. For example, the construction and maintenance of water conservation facilities such as canals, required for agricultural irrigation, were carried out through collective action with shared responsibility [15]. After farmland was put under scale management by a single operator, all the infrastructure required for scale management was needed to be independently built by the operator. On the other hand, the scale management operator also needed to make special investments in production equipment, such as purchasing large machinery, which required the investment of a large amount of capital. Moreover, agricultural production presents seasonal characteristics, and the highly expensive special machinery can be used only in a limited manner on the same piece of farmland, resulting in extremely high sunk costs.

The third is the extremely high supervision costs. It is extremely difficult to monitor the agricultural production process. In the period when farmland management was based on scattered families, the output of farmland was related to a household's livelihood and survival, and family members were naturally doing their best to cultivate farmland. Thus, there was no need to ensure a high supervision of the agriculture production process. In contrast, in the case of L Village, after the simple trading of the farmland use rights to farmers who operated large-scale farmland and a change in the pattern of the farmland managed by families, it was difficult for large-scale farmland operators to overcome the problems by strengthening their supervision in agricultural production. Specifically, when farmland is managed at a large scale by a single operator, the specific farmland production and management process have to be implemented by hired laborers. However, the wage settlement for hired laborers often precedes the farmland harvesting time; therefore, in the process of hiring laborers for agricultural production and management, since the current outcomes of labor need to be assessed until the harvest season, it is difficult to monitor and evaluate the current labor of all hired laborers, which brings extremely high risks to farmland scale management.

*4.2. The Common Use of Farmland by Different Users Allows Sharing the Costs of Farmland Scale Management*

The basic farmland trusteeship model, shown in Figure 1, and the farmland trusteeship model led by the Party branch, shown in Figure 2, share a common feature for farmland scale management; i.e., they reflect the common use of farmland by different users. Farmers need to obtain agricultural products and agricultural income through farmland to maintain their livelihoods and guarantee their survival. Simultaneously, the village collective needs to implement village governance through the management of farmland; and providers of productive services need to make profits from farmland. Therefore, it is the common use of farmland by different users according to their needs that allows a reasonable sharing of the three major costs arising from farmland scale management (Table 1), which are described below.

First, the supervision cost is borne by farmers themselves. The use rights of farmland are not actually traded to others when farmers participate in the farmland trusteeship, which indicates that the profits from farmland management finally belong to the farmers. As a result, the farmers' use of farmland is the most reliable way to make ends meet and guarantee their survival. Accordingly, in the context of farmland trusteeship, farmers

carefully supervise other operators who work on their farmland to ensure that their own livelihood and survival are not threatened. This solves the problem of the difficulty to ensure the effective supervision of farmland scale management.

Second, the coordination cost is borne by the village collective. In China, as the legal owner of farmland, the village collective must coordinate the common use of farmland in order to achieve the orderly management of assets, thereby clarifying the status of the landowners. Consequently, the implementation of coordination and management work on the farmland enables the village collective to form legal relations with each villager in production and life inside village and also to establish economic relations with various market organizations and social groups from outside. Therefore, the village collective becomes a bridge to coordinate various farmland users, thereby effectively settling the problem of coordination in farmland scale management.

Third, various market participants providing productive services share the production costs. In general, the costs of grain cultivation include the purchase of seeds, fertilizer, pesticides, and other agricultural materials as well as machinery and equipment investments and labor costs. On the one hand, the unified procurement by agricultural service organizations can ensure lower prices for agricultural materials more easily than those by individual farmers. On the other hand, market participants have already been engaging in specialized productive service work for a long time; therefore, they already own expensive production equipment through special investments. Our investigations clearly indicate that these organizations have purchased agricultural implements such as seeders, soil testing and fertilizer mixing machines, self-propelled sprayers, grain dryers, and drying towers, which are often unaffordable to individual farmers. Moreover, the mechanized operations are carried out uniformly on the concentrated big size farmland, thereby leading to significant reduction in the labor costs. Consequently, the utilization rate of machinery and equipment, as well as the production efficiency of farmland, is improved, and the average cost of equipment is reduced. In this way, the benefits of farmland scale management are achieved, and the problem of high production costs in farmland scale management is effectively solved.

**Table 1.** Common use of land and allocation of operating costs in socialized farmland management.

| Users | Purpose of Farmland Use | Shared Costs |
|---|---|---|
| Farmers | Gain agricultural income, maintain livelihoods, and guarantee survival | Supervision costs |
| Village collective | Gain political reputation and ensure village governance | Coordination costs |
| Service organizations | Gain profits and maintain management | Production costs |

The above-mentioned analysis clearly makes us understand why L Village did not successfully promote the farmland scale management in the end. In L Village, the entire 113.33 hectares of land was transferred to a large farmer household. Villagers have leased their land to others; therefore, they no longer enjoy the benefits of land management, and no longer care about the harvest. At the same time, the operating area is too large; thus, the large-scale farmer must hire labor to carry out agricultural production. However, in the process of employing labor force for agricultural production, the large-scale farmer is unable to carefully supervise agricultural production on site, which is prone to the problem of low quality of employed labor force that seriously affects the output of land in the harvest season. This study reflects that in the absence of household management, the large-scale farmer simply cannot afford the supervision costs of agricultural production. At the same time, the large-scale farmer needs to invest a lot of agricultural resources, machinery and equipment, and labor costs to operate large areas of land, which requires tremendous capital for maintenance. It indicates that the large-scale farmer has to bear high agricultural production costs. Therefore, under the pressure of high production cost and supervision cost, the large-scale farmer has to redivide the land into 10 pieces and entrust others to manage, in order to reduce the supervision and production cost. In this situation,

farmland management has returned to the previous state of decentralized management, and it ultimately cannot achieve farmland scale management. Clearly, compared with the cases illustrated by Figures 1 and 2, L Village did not realize the reasonable sharing of agricultural production costs and supervision costs, and all costs were solely borne by the large-scale farmer himself, which ultimately made it difficult to realize farmland scale management.

### 4.3. Joint Household Management Is the Basis to Promote Farmland Scale Management

The experience of FURT implemented in L Village indicates that farmland managed by farmer households can overcome the supervision problem in agricultural production, which makes farmland management face extremely high risks once disjointed from households. The case of scale management based on cooperation and negotiation in XSD Village and the shareholding of farmland management rights in XXW Village allowed us to find that the achievement of effective farmland scale management is not simply due to the concentration of farmland use rights but the joint management achieved based on household; it is herein named as joint household management.

In particular, in XSD Village, the farmland management rights are directly left to the farmer households, and the households' income from planting is directly calculated according to the output and the market price. In relation to the case of XXW Village, the shareholding of farmland use rights still guarantees that the output of the farmland is directly linked to their income. In other words, although farmland management is carried out through a joint stock partnership, farmers have an incentive to supervise agricultural production, because the yield of their land is closely related to their income. During the busy farming season, the most common sight in XXW Village and XSD Village is that family members stand beside their fields, supervising the operations of agricultural machinery across the farmland. Moreover, from time to time, they communicate with the agricultural machinery operators with regard to some unsatisfactory aspects and ensure the cultivation quality of the farmland, thereby protecting their agricultural yields and income.

The results indicate that farmland scale management in XSD Village and XXW Village is performed on the premise of ensuring household management based on households. Through such a farmland management pattern, mechanized agricultural production can be realized by purchasing agricultural productive services, which not only is time-saving and efficient but also can entail high crop yields. To ensure agricultural production, household members need only to return to their village and spend a little time supervising some essential farming links, while they can safely employ the rest of their time to work outside the village and obtain off-farm working income. Thus, farmland scale management does not necessarily need to be realized through the FURT. In fact, joint household management is the basis for effective farmland scale management under the premise of decentralized farm land property rights.

### 4.4. A Good Level of Collective Action Is the Premise to Realize Farmland Scale Management

At this point, a question further emerges: how to realize joint household management. If it is not possible to gather those farmers who prefer to work alone and conduct management in a decentralized way, then the unified agricultural productive services cannot be provided, and farmland scale management cannot be achieved. Therefore, to realize joint household management, it is necessary to gather the scattered households and effectively connect them to the market. It indicates that the formation of effective collective action around agricultural production in villages is crucial. Farmland scale management depends on not only the unified decisions made by farmers in agricultural production but also the concerted collective action after the unified decision making. On the one hand, through the leadership and organization of village leaders, households uniformly participate in the farmland trusteeship project, and the ideological collective action lays the foundation for farmland scale management. On the other hand, the legal, social, and economic relationships established among leaders, village collectives, farmers, and various social and

market organizations allow for the promotion of collective action on farmland production and management among different farmland users, ultimately ensuring the common use of farmland.

In the cases of XSD Village and XXW Village, the village Party branch plays a key role in organizing farmers to realize joint household management. In XXW Village, the most important role played by the Party branch in organizing farmers to invest in farmland use rights does not lie in realizing the concentration and trading of farmland use rights or in achieving reasonable profit distribution but rather in enabling farmers to form organized and unified behaviors regarding farmland scale management through the mechanism of profit distribution. Comparatively, in XSD Village, the pattern of organizing farmers is more intuitive, and it is primitively manifested as the coordination and mobilization of farmers by the village committee. Owing to the introduction of excellent seed resources, farmers must uniformly sow wheat seeds of the same type; and the large-scale benefits, such as agricultural productive services and company orders brought by such large-scale planting, make farmers realize the importance of mutual cooperation and unified action in production. As a result, in the long process of crop growth, although the FURT was not performed in the village, under the coordination and organization of the Party branch and in order to ensure cooperation in production, farmers adopted the practice of mutual supervision in seed selection, planting, mechanical operation, and harvesting. Consequently, in XSD Village, there was no behavior aimed at damaging cooperation, such as secretly hoarding excellent seeds or replacing them with others. Therefore, under the organization of the Party branch, the farmers of XSD Village achieved farmland scale management by promoting cooperation among all users during the common use of land.

In contrast, L Village chose to transfer all the land of the village to a large-scale farmer, which indicates that all villagers completely transferred the right of use of their land to the large farmer household, which broke away from the basis of household management, and small-scale farmers were excluded from agricultural production. Simultaneously, all agricultural production costs and risks were borne by the large-scale farmer alone. After the unified transfer of land use rights through the villagers' meeting consultation, the village collective no longer interfered with the operation of the large-scale farmers. Therefore, villagers and village collectives were excluded from the land, the collective action ability of villages gradually declined, and the success rate of farmland scale management became weak, and no result of farmland scale management was finally attained.

Thus, the results indicate that farmland scale management is inseparable not only from the foundation of farmland management based on households but also from the promotion effect of the collective action capacity in the village. Therefore, the key to realizing farmland scale management is to encourage the scattered households to cooperate with one another through an effective organization, so that the originally fragmented villages and the markets can effectively cooperate and connect, finally realizing the common use of farmland by different users.

*4.5. The Land Trusteeship System Satisfies the Essential Requirements of Socialized Farmland Operation*

In accordance with Marxism plutonomy principle, the socialized operation or production of a resource is embodied in three basic processes: the socialization of resource use, the socialization of resource production process and the socialization of resource output [35]. Figure 3 summarizes the general process of farmland scale management in the Chinese context. Clearly, the farmland scale management satisfies the three major requirements in the process of socialized farmland operation.

The first requirement is the socialization of farmland use, which refers to the process of change from a scattered use of farmland by individuals to its common use by multiple users. Figures 1 and 2 and Table 1 together present that in the process of farmlands use by different users, farmland has actually become the source for farmers to maintain their livelihood, the starting point for the village collective to implement effective governance,

and the way for service organizations to gain profits. Therefore, the process of common use of farmland allows for the formation of the socialization of farmland use.

The second requirement is the socialization of the farmland management process, which is the process of change in farmland management from a series of individual actions to a series of social actions. Therefore, the process in which the farmers entrust different operating links to service organizations for management is actually manifested as the process of agricultural production from the original farmers to undertake all production links independently compared to the current process of different service organizations to undertake different production links. This is the socialization of the farmland management process.

The third requirement is the socialization of farmland output, which indicates that the output of farmland is cooperatively enjoyed by different users participating in farmland use through the joint use of farmland and reciprocal coordination and cooperation. Based on joint household management, the products produced by the farmland are divided among the farmers. In parallel, based on the principal-agent relationship, the partial profits from farmland output are divided among various organizations that provide productive services. However, based on ownership relationships, the political gains from coordinating farmland management are enjoyed by the village collective.

Combined analysis of these three processes (i.e., socialization of farmland use, socialization of farmland management process, and socialization of farmland output) indicates that in the context of the Chinese institution, the essence of farmland scale management is actually manifested through the socialized operation of farmland. Therefore, the process of realizing farmland scale management corresponds to the process of socialization of farmland use, socialization of farmland management process, and socialization of farmland output required by socialized farmland operation.

Based on the aforementioned analysis of the connotation and characteristics of socialized farmland operation, a comparative analysis was made in this study between two types of farmland scale management: the one based on FURT and the other based on farmland trusteeship. It was found that in farmland scale management based on FURT, it was impossible for an individual operator to solve the problem of high costs. This is also the reason why FURT is mired in difficulties. In contrast, the institutional model of farmland trusteeship allows for an effective sharing of the costs based on the common use of farmland by different users.

In fact, FURT and farmland trusteeship can be considered as two different stages in realizing farmland scale management. The case represented in Figure 2 indicates that FURT can be considered as an early stage of this process, addressing the issue of how to reorganize the farmland users according to the actual local conditions. Therefore, the simple reorganization of farmland users cannot satisfy the three above-mentioned requirements of socialized farmland operation. In fact, in order to truly realize the farmland scale management, it is necessary to form a systematic institution that can meet the requirements of socialization of farmland use, socialization of farmland management process, and socialization of farmland output, such as the institution of farmland trusteeship. Therefore, the case represented in Figure 1 more intuitively shows that the socialized farmland operation that fulfills the three requirements of socialization is the very essence of farmland scale management in China's institutional context.

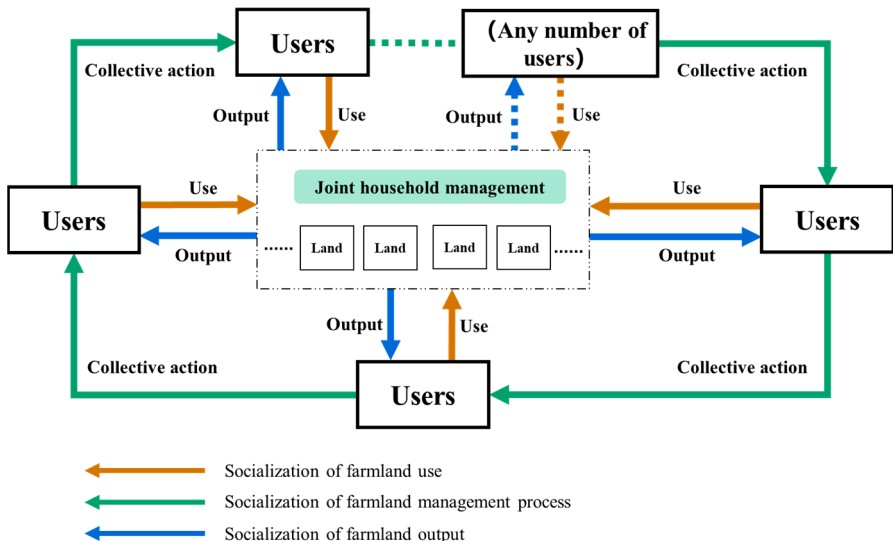

**Figure 3.** Conceptual framework of socialized farmland operation.

### 4.6. Challenges to the Promotion of Socialized Farmland Operation

Although farmland scale management can be achieved through socialized farmland operation under the background of fragmented property rights and decentralized management in China, its development in practice is hindered by several factors.

In fact, irrigation problems seriously restrict the development of scale management. In particular, in XXW Village, although the farmland joint-stock cooperative led by the Party branch achieved good results, the question aroused as to why only about 20% of the village's farmland joined the farmland joint-stock cooperative. To answer this question, the Village Party Secretary of XXY Village pointed out that it was the irrigation water that could not be guaranteed. In other words, limitations in water conservancy facilities and irrigation water conditions are the biggest problems affecting the production expansion in joint household management. Problems such as decline of collective irrigation, shortage of groundwater resources due to the drilling of wells, high irrigation costs, and declining river water quality hindered the development of joint household management. In contrast, XSD Village, located within the spatial coverage of the pumping stations of the Water Conservancy Bureau, has good water conservancy conditions, which favors the realization of joint household management.

Moreover, farmers are worried about whether the leadership of the village can be continued. In our investigation, several villagers pointed out that although the current farmland scale management model of joint farmland management formed under the leadership of the Party branch was very good, they were tremendously worried about the following problems: In the future, if the superior leadership will change or the village committee members will be replaced, will the current form of organized farmland trusteeship and joint household management still be guaranteed? Will there be anyone still organizing it? Therefore, the uncertain predictions of the future were also the reason why some farmers were reluctant to accept the pooling of farmland as shares, even if they had such an opportunity or joined cooperatives organized by the local village committee.

### 5. Data Evidence: The Development and Effect of Socialized Farmland Operation in China

The public statistics data indicate that China is gradually realizing the process of farmland scale management; in fact, it is a process of gradual realization of the socialized farmland operation.

China's farmland scale management has experienced the process from land fragmentation and FURT and then the transition to farmland trusteeship. Owing to the extremely scattered and fragmented land property rights of farmers, coupled with the outflow of

agricultural labor force and other factors, China has experienced the process of small-scale farming in a long period of agricultural development. Most of these methods are accompanied by higher agricultural production costs, inconvenient agricultural management, and low farm profits and efficiency [36]. According to *the third National Land Survey*, China's per capita arable land area was only 0.09 hectares at the end of 2019, which is less than 40% of the average level over the entire world. In response to the challenges faced by agricultural production, the Chinese government encourages the farmland scale management through FURT. Data show that China's FURT rate has increased year by year, from 5.2% in 2007 to 35.1% in 2016 [14]. According to the Ministry of Agriculture and Rural Affairs, PRC (2021), by the end of 2020, the FURT area of the country has reached 35.47 million hectares. Among them, a particular concern is that in recent years, the growth rate of China's FURT rate continues to decline, and the ratio of FURT has remained at about 40% [37]. Existing studies have not reached consensus explanation for this statistical result. However, according to our research, this is actually the result of not realizing the socialized farmland operation, which leads to the high cost of agricultural production and limits the farmland scale management to a certain extent. The state of L Village is actually an epitome of such a situation.

In contrast, land trusteeship has developed particularly rapidly in recent years and has been highly valued by the central government and local governments as well as welcomed by farmers. On the whole, the area of land trusteeship in China shows an increasing trend year by year. Data show that in 2016, the land trusteeship area expanded to 28 provinces in China, with a trusteeship area of 6.67 million hectares (data from the official website of all China federation of supply and marketing cooperatives). By the end of 2021, China's land trusteeship area exceeded 111 million hectares, supporting more than 78 million small-scale farmers. In our study area, land trusteeship in Shandong Province increased significantly from 0.35 million hectares in 2014 to 2.43 million hectares in 2019. By 2022, Shandong's land trusteeship area would reach 8 million hectares.

The data corresponding to the process of rapid development of land trusteeship clearly reflect the three core requirements of socialized farmland operation. First, in terms of the socialization of farmland use, by 2022, the number of agricultural socialization service organizations in Shandong had reached 122,000, and the number of agricultural cooperatives had reached 244,000, serving 13.7 million farmer households. By the end of 2022, there are more than 2300 villages in Shandong Province carrying out land trusteeship, more than 6000 cooperatives run by Party branches, and more than 1100 village officials participating in land trusteeship. In our case village, the land trusteeship of XSD Village has developed from a land trusteeship covering one village to a land trusteeship covering a lot of villages, forming a community to carry out socialized farmland operation. By September 2021, more than 2000 farmers in the township where XSD Village is located have entrusted about 530 hectares of farmland to the agricultural service company. Meanwhile, the land trusteeship area of XXW Village has grown from 25.8 hectares to 86.67 hectares in 2023, and the number of participating households has grown from 102 to 212. The results clearly reveal that land trusteeship has driven many farmers, village collectives, cooperatives, and other market entities to participate in farmland management, showing significant features of socialization of farmland use.

Second, in terms of socialization of the production process, according to incomplete statistics, since 2022, 56,000 service organizations in Shandong Province have provided productive services for wheat production to 6.69 million farmers. Moreover, 215,066.67 hectares of land were entrusted to service organizations for all wheat production links, and 1.4 million hectares of land were entrusted to service organizations for multiple links in wheat production. In these processes, agricultural companies have purchased large machinery and equipment to provide trusteeship services for villagers. Among them, the agricultural company is responsible for building a professional team and hiring agricultural experts for technical guidance in order to achieve scientific fertilization and pesticide use. Local farm experts and village cadres are also hired as field managers, who are responsible for field management, pest monitoring, crop management, and coordination with farmers. In our case village, the farmland of XSD

Village is managed uniformly by the village collective, and the agricultural service company is entrusted to the integrated land. By 2022, the agricultural service company had built 30 sets of mechanical equipment, including drying towers, drying yards and grain warehouses. Thus, the traditional process in which a single farmer household is in charge of all agricultural production links has been transformed into a process jointly completed by different operators in the process of gradually promoting land trusteeship.

Third, in terms of the socialization of farmland output, in land trusteeship, land output is shared by relevant participants involved in land use and management. For example, in Xiajin County, Dezhou City, Shandong Province, cooperatives run by Party branches cooperate with agricultural companies to operate cultivated land. After deducting all planting costs, the surplus is shared out in a ratio of 4:3:3 among agricultural companies, village collectives, and farmers. In the city of Jinan, Shandong Province, agricultural service companies sign land trust agreements with farmers, with villagers receiving 1771 US dollars per hectares of land and village collectives receiving 66.41 US dollars per hectares of land. After the sale of grain profits, villagers participate in the dividend. In the city of Shenzhou, Hebei Province, 154 farmers entrusted more than 200 hectares of land to a specialized agricultural machinery cooperative, with 60% of the sales income going to the farmers and 40% going to the cooperative. In our case village, in the first half of 2022, farmers participating in land trusteeship in the township where XSD Village is located received a guaranteed return of 17,780 US dollars per hectare and a surplus dividend of about 1112.25 US dollars. After the implementation of land trusteeship in XXW Village, the farmers and village collective share the income from farming. In recent years, farmers in XXW Village have added an average of nearly 300 US dollars a year to their income through land trusteeship. All the above-mentioned examples show that the land output is shared by all land users and reflects the socialization of land output.

From the perspective of benefit, considering Shandong as an example, through the socialized farmland operation, represented by land trusteeship, the planted area could be increased by more than 5%. With the use of large modern harvesters, the wheat loss rate was reduced to less than 1%. Moreover, unmanned plant protection aircraft and self-propelled spray machine were adopted, which are more than 40 times more efficient than manual spraying by farmers. The agricultural company provides full productive service for 512.93 hectares of land in 12 villages in three towns in Xiajin County. In 2022, these 12 villages received a total of 820,508.40 US dollars, among which the village collective income increased by more than 177,120 US dollars. In our case village, the township where XSD Village is located will produce 7500 kg of wheat per hectare after the autumn harvest in 2022, and the annual average income of village collective will reach 37,075 US dollars. After the implementation of land trusteeship in XXW Village, the average input cost of agricultural resources per hectare of farmland has been reduced from 2058.2 US dollars to 1767.97 US dollars. At the same time, according to statistics, land integration through the consolidation of ditches and roads has increased by about 2%.

In a word, the analysis of the above-mentioned data indicates that with the continuous increase in land trusteeship, the socialization of land use, the socialization of land production process and the socialization of land output are also deepening simultaneously. This enables us to more intuitively observe through data analysis that the essence of the existing more popular farmland scale management mode is the socialized farmland operation represented by the socialization of land use, land production process, and land output. Compared with the farmland scale management achieved through FURT, the farmland scale management achieved through the socialized farmland operation is more conducive to the improvement of the scale operation level on the premise of not excluding small farmers.

## 6. Conclusions, Implications, and Prospect

### 6.1. Conclusions

Based on a case study analysis of Shandong Province, China, this study pointed out that the process of realizing farmland scale management through farmland trusteeship actually corresponds to the process of socialization of agricultural land use, socialization of farmland management process, and socialization of farmland output, thereby satisfying the requirements for socialized farmland operation. This is the essence of farmland scale management reflected in China's institutional context. Furthermore, socialized farmland operation does not exclude the existing FURT. In our opinion, FURT is the process of reorganizing the participants involved in socialized farmland operation, and it is an important step that can be selected to realize socialized farmland operation.

### 6.2. Theoretical Implications

How to successfully realize the farmland scale management in the countries with extremely dispersed land property rights has not only always been the research hotspot discussed by scholars in developing countries but also an important issue that has not reached conclusions with high practical values. Through the discussion of the process of socialized farmland operation, this study may contribute some new understanding in this aspect.

First, existing studies on how to achieve farmland scale management in countries with a large number of small and scattered farmers are mostly reflected in discussing the result when achieving farmland scale management [38–40]. However, few studies discuss the farmland scale management as a process. Therefore, many studies do not emphasize the interactive process of different land users in realizing farmland scale management. This study holds that the process of realizing farmland scale management is actually a process of collective action among different land users, which provides a more general logical way to better understand the problem of farmland scale management from the perspective of institutions.

Second, existing studies on how to achieve farmland scale management in countries with a large number of small and scattered farmers have put forward a large number of practical pathways to realize farmland scale management [41–45]. These pathways explain how to achieve farmland scale management in specific places, but they are hardly universally applicable. Based on the theory of collective action and the path of land trusteeship, this study summarizes that the essence of farmland scale management is the process of socialized farmland operation. In fact, it presents the general laws and principles of farmland scale management through the concrete practice of land trusteeship.

Third, the above-mentioned discussion further reveals that based on the phenomenon, farmland scale management is the continuous expansion of the scale of farmland resources. However, in essence, the connotation reflected by farmland scale management is the interaction between the ecological systems represented by land resources and the social systems represented by land users. Therefore, the farmland scale management is actually the result of optimization of the internal interaction of a social–ecological system with land as the core.

Finally, from the perspective of collective action theory, existing studies on collective action mostly discuss the role of collective action in resource and environmental management [46–48]. Based on the analysis of the process of common use of farmland by different users, this study further reflects how collective action meets the requirements of advanced productivity at the micro-level, and it further expands the theoretical significance of research on collective action.

### 6.3. Political Implications

Currently, COVID-19 is impacting food security by affecting agricultural production and food availability. This study explores how to cope with the high cost of agricultural production and the decline in comparative benefits of agriculture through cooperation

under the impact of COVID-19, thus providing policy implications for developing countries, in particular, those with large number of small and scattered farmers.

First, this study provides new practical ideas for coping up with the decline of food production capacity under the epidemic. The epidemic affects the input of agricultural labor force and related production factors, resulting in production disruptions. Epidemic prevention measures in some countries have restricted the movement of workers in the food and agriculture industry, which to some extent aggravated the shortage of agricultural labor force and the difficulty of purchasing agricultural inputs needed for food production, thus delaying the farming season and affecting food production. For food-importing countries, supporting domestic agricultural production and expanding import sources can help stabilize food supply during the COVID-19 pandemic. However, in the long run, a focus on increasing agricultural production and reducing import dependence is more important for countries to become more self-sufficient in food. Moreover, this study presents a new way of agricultural farmland management, in terms of improving agricultural productivity, and the increase in food self-sufficiency plays an important role. Through the socialized farmland operation, the lowest cost can be exchanged for the highest efficiency; thus, it can well deal with the problem of global reduction in agricultural production and production disruption caused by COVID-19.

Second, this study provides implications for ensuring the stability of the food supply chain under the epidemic. COVID-19 has disrupted the stability of the global food supply chain. FAO experts believe that the COVID-19 pandemic could lead to nearly 90 million additional food shortages worldwide, in particular, in developing countries with low agricultural productivity and poor infrastructure [49]. Unfortunately, the disruption of the agricultural supply chain caused by COVID-19 has made global food security more severe. The increased cost caused by transport delays is expected to be eventually reflected in the consumers aspect, resulting in higher food prices and affecting the ability of low-income groups to buy food. The new agricultural operation mode presented in this study can play an important role in ensuring the effective supply of agricultural products, stabilizing domestic food production, maintaining agricultural production order, maintaining food supply and price stability, and thus maintaining the stable development of society and economy.

Third, this study provides implications for how to deal with the impact of the epidemic through cooperation. COVID-19 affects food security and agricultural development. In the context of COVID-19, the theme of global development is to achieve cooperation and to work together to cope with the shocks and challenges of unexpected scenarios. Thus, how to work together has been a question that people have been talking about in the context of COVID-19. This study explores how to develop a form of joint operation in the field of agricultural production under the impact of environmental shocks, such as labor outflow and modernization, thus providing a new idea for countries around the world, in particular, developing countries to deal with the impact of the epidemic through cooperation.

*6.4. Prospect*

In China, the promotion of land trusteeship and further advancement of the socialized farmland operation face significant challenges. Therefore, it is necessary to further strengthen the cultivation of rural organization, leadership, and other aspects, so as to further promote the farmland scale management.

First, the limitation of natural resources on agricultural production should be overcome to promote the socialized farmland operation. Considering irrigation water as an example, based on the fact that farmers can have organized agricultural production cooperation, it is necessary to strengthen the input and management of irrigation water and actively introduce large-scale water-saving irrigation facilities for large-scale agricultural production, such as self-propelled sprinkler irrigation systems, to improve the efficiency and effectiveness of irrigation. In areas where large-scale farmland concentration and cooperative production can be created, collective irrigation facilities and institutions should

be actively repaired, and the entire management of collective irrigation should be innovated based on the saving of irrigation costs and protection of groundwater resources from over-exploitation.

The second objective is the continuous promotion of the construction of high standard basic farmland (high-standard basic farmland refers to the concentrated contiguous farmland formed through land integration, with supporting facilities, high and stable yield, good ecology, strong disaster resistance, and suitable for modern agricultural production and management mode) and to improve the infrastructure of agricultural production. Land consolidation is a process that transforms dispersed lands into a complete unit. Land management has more economic reasons under the premise of concentration of cultivated land [50]. In the future, land utilization structure and layout should be optimized through land consolidation, soil improvement, field road, and other infrastructure construction, and farmland infrastructure conditions should be constantly improved to provide convenient conditions for the development of farmland scale management.

Third, it is necessary to strengthen the stability of village leadership, and the organizational foundations of socialized farmland operation should be consolidated. The way of dispatching leaders and cadres to villages is a supplement to the villages' leadership, and it is an effective approach to guarantee the formation of their good organizational ability and collective action ability. In the future, rural areas should continue to explore effective ways to cultivate local leadership and introduce external leadership, as well as to stabilize the rotation mechanism of village branches and village-level leaders, in order to stabilize farmers' expectations over the organizational ability and collective action capacity of villages.

Finally, importance should be attached to the cultivation of the organizational ability and collective action ability of villages, so as to provide organizational motivation for socialized farmland operation. In this respect, it is necessary to strengthen grassroots governance capacity and public service system construction; actively develop professional services and rural cooperative mechanisms; and vigorously develop rural e-commerce and e-government services. Moreover, it is further highly desirable to attach importance to, and strengthen, rural cultural construction and the building of democratic rule of law; deepen the institutional reform of rural collective property rights; and promote institutional innovation according to local conditions. In this way, it would be possible to comprehensively improve the organizational and collective action ability of the villages, respect farmers' practical innovation and local pilot experiments, and explore institutional arrangements that suit local characteristics, with the objective of enhancing rural collective action capacity and supporting the realization of farmland scale management.

**Author Contributions:** Methodology, Q.H. and L.Z.; Formal analysis, Q.H.; Investigation, Q.H.; Resources, Y.S.; Writing—review & editing, H.X.; Supervision, Q.M. All authors have read and agreed to the published version of the manuscript.

**Funding:** This research was supported by the National Social Science Foundation (Grant No. 22GBL225).

**Institutional Review Board Statement:** Not applicable.

**Informed Consent Statement:** Not applicable.

**Data Availability Statement:** Not applicable.

**Conflicts of Interest:** The authors declare no conflict of interest.

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
