# Peer review of "Socialized Farmland Operation—An Institutional Interpretation of Farmland Scale Management"

_sustainability, doi:10.3390/su15043818_

Round 1

Reviewer 1 Report

Taking the farmland trusteeship practice implemented in Shandong Province of China as the research case, this paper discussed the essence and realization premise of the new farmland scale management model represented by farmland trusteeship based on case analysis and obtained some conclusions. The topic selection of this paper has some significance and interesting, the analysis is more reasonable, and the conclusions are more valuable. However, there are still some problems:

1. This paper mainly analyses the effect of farmland scale management qualitatively, but there is no quantitative data. It affects the credibility of the conclusion. Quantitative evidence needs to be supplemented.

2. This paper mainly analyzes the scale management of farmland in typical villages, and does not compare and analyze the difference between the implementation and non-implementation of scale management of farmland. It is recommended to supplement.

3. The paper considers that irrigation condition is an important factor affecting farmland scale management. Are field roads and other infrastructure also important factors affecting scale management of farmland?

4. The title of 4.4 (line 465) and 4.5 (line 509) are the same. It is needed to check and modify.

5. The figure 2 (line 317) and the figure 3 (line 561) are the same. It is needed to check and modify.

Reviewer 2 Report

I tried to read the manuscript “Socialized Farmland Operation — A Institutional Interpretation of Farmland Scale Management”.  The manuscript examines the trading of farmland use rights in China.

The issue is arguably interesting though the writing style is problematic (needs a careful proof-reading and editing). However, I do struggle with how to categorize the manuscript.It is not an empirical exercise, a conceptual contribution that reorganizes complex issues, or a thorough literature review.   

There is a lot of work to be done in order the authors to have a second chance.

Just to mention some:

1) Title : What is “socialized farmland operation” ? What is “farmland scale management”?

2) citation [4] has nothing to do with COVID.

3) Lines 55-60 : confusing sentences.

4) Line 72 : What is “large-scale productive services for rural households”?

5) Line 92: what is : “from the perspective of farmer differentiation”?

6) Line107  [26] irrelevant citation

7) Line 112 “principal-agent issues”

8) Line 113 what is the “personification of property rights”?

9) Line 140-141 The “ the number of members of farmers who are involved in non-agricultural employment” is a not accurate concept. A member cannot be involved in non-agricultural employment. Presumably the authors want to say “off-farm employment”

10) Line 142-143 I have no idea what is the “ the optimization degree of the farmland trusteeship market”.

Round 2

Reviewer 1 Report

The author has modified and supplemented the paper, basically meeting the requirements, but there are still the following problems:

1. Using quantitative data to verify the effect of agricultural land scale management can not only use the data of the whole country and Shandong Province, but also supplement the data of two cases.

2. The positions of Lines in part of the revision instructions do not correspond to the revised version, for example, the revised version lines 669-760 in question 1; Some cannot be found in the corresponding position, such as the revised draft Lines 870-877 in question 3, which needs to be checked again.
